# Recent Evidence on Polycyclic Aromatic Hydrocarbon Exposure

**DOI:** 10.3390/healthcare11131958

**Published:** 2023-07-07

**Authors:** Xiaohan Zhao, Jiuhe Gao, Lingzi Zhai, Xi Yu, Ying Xiao

**Affiliations:** 1State Key Laboratory of Quality Research in Chinese Medicine, Macau University of Science and Technology, Taipa, Macao 999078, China; xiaohanzhao-jacqueline@outlook.com; 2Faculty of Medicine, Macau University of Science and Technology, Avenida Wai Long Taipa, Macau 999078, China; jiuhegao000@gmail.com (J.G.); lingzizhai@outlook.com (L.Z.); 3Guangdong-Hong Kong-Macau Joint Laboratory for Contaminants Exposure and Health, Guangzhou 510006, China

**Keywords:** dietary pattern, polycyclic aromatic hydrocarbons, pollutant exposure

## Abstract

This review provides a comprehensive conclusion of the relationship between the intake of various polycyclic aromatic hydrocarbons (PAHs) and different dietary patterns, pointing to the accompanying potential health risks. To achieve this, existing pertinent research was collected and analyzed. The collation revealed that the concentration of PAHs in food and their dietary patterns were diverse in different regions. Specifically, the concentration of PAHs in food was found to be related to the level of pollution in the area, including soil, air, and water pollution, which is then accumulated through the food chain into food that can be ingested directly by the human body, resulting in malformations in offspring, increased risk of cancer, and gene mutation. Guidebooks and dietary surveys were consulted to uncover disparities in dietary patterns, which indicated regional variations in taste preferences, traditional foods, and eating habits. Different regions are spatially categorized in this assessment by cities, countries, and continents. Notably, smoking and grilling are two of the food processing methods most likely to produce high levels of PAHs. To prevent excessive intake of PAHs from food items and attain a higher quality of life, more health education is urgently needed to promote healthy eating patterns.

## 1. Introduction

Polycyclic aromatic hydrocarbons (PAHs) are a category of hazardous chemical compounds that contain two or more benzene rings with more than 200 different types. They can be formed through incomplete combustion of organic waste or pyrolysis of organic substances [1]. PAHs possess a unique structure that provides them with thermal stability and recalcitrant to degrade [2,3], allowing them to remain in the environment and human body with barely any metabolization [4]. Additionally, their hydrophobic nature makes them more likely to accumulate in soil and contaminate crops. PAHs have been widely recognized as carcinogenic, teratogenic, and mutagenic. Except for that, Alzheimer’s disease (AD), cardiovascular diseases (CVD), and chronic obstructive pulmonary disease (COPD) are also considered to be consequences of overexposure to PAHs [5,6,7]. Since 2007, the upper limit of benzo[a]pyrene (B[a]p) in food has been governed by Regulation (EC) No. 333/2007, on behalf of the beginning of more attention being paid by people to the hazards of PAHs. Benzo[a]pyrene is a typical PAH that has a high level of toxicity and is ubiquitous in nature. In 2011, an updated addendum to the term stipulated that the sum of four different kinds of PAHs, benzo[a]anthracene, chrysene, benzo[b]fluoranthene, and benzo[a]pyrene (PAH4), were to replaced benzo[a]pyrene as new marker substances. This move has represented a critical step in observing and monitoring the adverse effects of PAHs on human health and the environment by relevant organizations.

PAHs have been detected in the air, food, polluted water sources, and soil. The route of human exposure to PAHs includes skin contact, inhalation, and ingestion, with dietary intake as the predominant pathway for most nonoccupationally exposed individuals. PAHs can be found in a variety of common foods such as meat, dairy products, nuts, vegetables, fermented foods, smoked foods, etc. From the perspective of food ingredients, unavoidable PAHs are likely entering the human body through the pollution of atmosphere and soil, which would affect roots and the internal circulation of plants and animals while accumulating in the food chain. Extra PAHs may be formed during the production process of cooked foods when compared to raw and cold ones, particularly when smoking, frying, and grilling. As shown in Figure 1, PAHs enter the population of various ages according to the dietary habits of different regions. According to an article published in Sweden in 2010, the B[a]p content of samples of smoked meat ranges between 8.4 and 14.4 μg/kg [8], but the B[a]p content of raw fish is less than 0.03 μg/kg [9]. Based on the average of the two studies conducted between 1999 and 2010, the B[a]p content of smoked meat ranged from 6.6 to 36.9 μg/kg, whereas the B[a]p content of raw meat was 0.04 μg/kg.

From another perspective, the decline in human quality of life (QOL) resulting from the consumption of PAHs is an issue that cannot be disregarded. Take the most ubiquitous disease, cancer, also known as malignant tumor; it is a kind of long-term disease of abnormal cell division and proliferation whose etiology is not completely clear. It is estimated that, in 2020, there will be approximately 19.3 million new cancer diagnoses, resulting in nearly 10 million fatalities. Despite advances in treatment, there remain several hurdles to overcome, such as the high cost of targeted therapies, adverse effects associated with chemotherapy, and disease recurrence. According to a publication by the World Health Organization (WHO), B[a]p had been justified as having a definite carcinogenic effect and categorized in Group 1 carcinogens [10]. Furthermore, the remaining PAHs are also proven to be risky in inducing the risk of cancer, even though they are not yet listed by WHO. To quantify the impact of PAHs to human health outcomes, the Incremental Lifetime Cancer Risk (ILCR) model, as proposed by the United States Environmental Protection Agency (U.S. EPA), is utilized as a valuable tool for calculating and evaluating health risks. A noteworthy correlation exists between health risk and daily PAH intake in distinct regions, as evidenced by disparities in ILCR values and toxic equivalents.

In order to distinguish different dietary patterns, countries and regions were used to break down the classification. As a whole, there are multiple eating patterns within continents or geographic directions as a result of divergent historical and cultural development. The income of different people may also play a role in PAH daily intake, since the food choices of residents will be influenced to some extent. As such, the content of PAHs in food varies between local diets. However, some popular dietary patterns lack geographic and cultural specificity, such as the Mediterranean diet being recommended as protective against cardiovascular disease without being representative of the region. Dietary Approaches to Stop Hypertension (DASH) is well known for its prevention and treatment of hypertension, but it can only be called a functional diet rather than a special diet with regional characteristics. Despite an increased amount of research linking food and PAHs, studies focusing on region-specific dietary patterns remain scarce.

This article reviewed the levels of PAHs found in global raw and processed foods, as well as the average daily intake of PAHs by residents and its impact on the risk of cancer. To understand the relationships between dietary practices, PAH intake, and to explore healthier lifestyles for residents in different areas, the dietary practices of different regions, their traditional characteristic diets, and daily dietary preferences were systematically reviewed and discussed. Meanwhile, risk factors in the process of food were discussed to better explain the connection between dietary patterns and cancer risk. The aim of this review is to present the latest evidence on dietary PAH intake and to indicate the relationship between dietary PAH intakes and cultural as well as dietary habits, while promoting healthy diets and improving the quality of life by proposing new concepts, recommending the types of food to be consumed according to the concentration of PAHs, and identifying initial habits that may increase the risk of cancer.

## 2. PAHs Contents in Regional Raw and Processed Foods 

Diverse cultures and dietary patterns also exist between macroscopic groupings. This phenomenon occurs in different countries on the same continent. In Asia, Indian cuisine is usually greasy and salty because residents there are likely to use stir-fry and exhibit preferences for spices and flavors; in contrast, the traditional Japanese food ‘sushi’ is known for its simple and bland taste [11,12]. As populous countries, there are also different eating habits within countries. For instance, as illustrated by the example of the Lingnan region in southeast China, the hot and humid climate leads residents to prefer light-tasting foods that are often prepared through boiling and steaming [13]. However, in east China, take Sichuan as an example, most individuals favor stir-fried dishes with a spicy and savory flavor profile, which reflects the regional characteristic [14]. It, thus, becomes arbitrary to categorize the preference of a particular culture by country. In order to solve this problem, data in this work was collected from further research to set standards for different preferences as the first step; then, the dietary survey was compared with the standard in different regions to get a relatively accurate conclusion.

In Table 1, studies on the content of PAHs in different local food items and their dietary pattern differences were summarized. In each set of data, quantities with excessively large differences were excluded to ensure uniformity and accuracy. Generally speaking, the average level of vegetables’ PAHs in Shanghai approached 205.1 ng/g. Compared with other cities, this eye-catching indicator showed a significant outburst. In terms of processed food, smoked fish in Sub-Saharan Africa was far ahead from others by approximately 310.1 ng/g. On the contrary, PAH_15_ concentrations in peanut, sugar, dry yam, dry cassava, broth cube, dehydrated milk, fresh tam, and fresh potato all remain below 1. In Spain, tubers and fruit also have PAH_15_ concentrations below 1. It is worth noting that ‘meat and meat products’ and ‘oil and fat’ are apparently higher than other concentrations in Spain. The same phenomenon happened between daikon and other detected vegetables in Shanghai, 2018.

Raw food are defined as foods which are uncooked and unprocessed [23]. For these food items, the regional differences and abnormal peaks of PAHs may relate to PAH concentrations in local soil. Romaine showed a peak of 320.6 ng/g in examined vegetables in the research by Jia and his colleagues [17]. Compared with vegetable PAH_total_ concentration in Shenzhen, China in 2013, the formal is approximately 16 times that of the latter [17,18]. Further investigation showed that the PAH_15_ in the woodland of Shanghai was about 1888 ± 552 ng/g, and the median and mean results were similar. Although the content of PAHs in the soil for planting has not been specifically detected, the results show that the nonurban land in Shanghai may contain relatively high levels of PAHs, and 4–5 ring PAHs are the main pollutants, such as fluoranthene(flu), chrysene (Chr), and Benzo[b]Fluoranthene (B[b]f), main sources of which were biomass burning and vehicles [24]. Meanwhile, the average value of PAH_28_ in woodland soil in Shenzhen, China was only 546 ng/g, while PAH_15_ in Shanghai was included in 28 detected PAHs [25]. Therefore, this group of data is of stark contrast. The average content of PAH in the soil of urban and rural areas in Beijing is 886 ng/g, and the content of PAH_15_ in vegetables is only 3.31 ng/g, confirming this conjecture again [26]. In addition to growing crops directly from the soil, it is also possible that polycyclic aromatic hydrocarbons can be enriched through the food chain. In the Spanish study, the PAH_16_ of meat, meat products, and oily foods were significantly higher than other indicators at 38.99 and 18.75 ng/g, respectively. In the soil test of Tarragona County, Spain, the average value of PAH_16_ was 1105 ng/g [27]. Combined with Spanish food culture, the content of PAHs in local meat and meat products has been super high since the survey in 2010 (179 g/person/day). According to research in Khana, Niger Delta region, where PAH_16_ in the industrial area approached 545–10,785 ng/g via diffusion into livestock feed and turf, PAH_16_ concentrations in chicken meat and cow meat were 16.7 ng/g and 25.4 ng/g, respectively, which were 2.6–4 times higher than fish (6.42 ng/g) [20]. Therefore, more meat intake also leads to a greater demand for feed and food, resulting in excessive PAH_16_ concentrations [28]. The same is true for oil and fat products.

The way a food is processed also plays a role in its PAHs concentration. As one of the oldest food processing methods, smoking and grilling can improve the preservation time of ingredients and enrich the flavor of foods. The documents published by the Codex Alimentarius Commission (CAC) and the Recommended International Code of Practice (RCP) list 10 factors that may affect the generation of PAHs: type of fuel, direct or indirect smoking or drying methods, the relationship between the smoke-generating process and pyrolysis temperature, distance between food and heat source, position of food in relation to the heat source, fat content of foods and changes in processing, duration of smoking and direct drying, temperature during smoking and direct drying, cleaning and maintenance of equipment, and the smoke density in the smoking chamber. Although there are different smoking and grilling processes, it is certain that many conditions cannot be controlled completely due to the limitation of techniques. Therefore, even smoked and grilled foods produced from the same batch may present divergences in PAH concentrations, let alone using different methods from different cultures and regions. According to research in the UK, it was found that the PAH concentrations of roasted bacon and crispy bacon were 1.75 and 1.08 ng/g, respectively [29]. Smoked sausage Slavonska kobasica was also detected as a popular traditional food in Croatia; ‘Traditional smoking in natural casing hanging 2 m from the fire’ is the traditional craft of making the precise sausage. Its PAH_16_ concentration reaches 679 ± 2.53 ng/g. However, in the group of ‘Industrial smoking in natural casing’, it is only 234 ± 3.48 ng/g in this regard [30]. Compared with roasted and crispy bacon in British cuisine, even the minimum PAH_16_ concentration in Slavonska kobasica was also 133–216 times higher than the former (approximately). For intragroup comparison, the PAH_16_ concentration of Slavonska kobasica in industrial production under controlled conditions was approximately 1/4 that of natural production (home-processed). In Spain, which is civilized by smoking and fermentation technology, the traditional fermented sausage Chorizo Gallego has the highest content of Phenanthrene and Naphthalene, RBaP/_Σ15PAHs_ = 0.77 ng/g, Σ15PAHs = 98.48 ng/g (dry weight) [31], significantly lower than Slavonska kobasica and higher than the two types of bacon tested in the UK. The results, the study showed, may be related to different smoke compositions and their deposition intensities.

## 3. Population Exposure of PAHs from Diet in Different Regions

To further investigate the correlation between PAH content in food and diet and the actual human exposure, Table 2 is used to visualize the per capita daily intake of PAHs by country or continent. As one of the riskiest human-known PAHs, B[a]p was set as a marker substance for PAHs according to the 2004/107/EC policy, published by the European Union (EU). It has uncontrollable destructive power over the human body; for instance, cytochrome P450 can produce carcinogenesis active metabolites [32]. Therefore, information about PAH daily intake (μg/day) was also organized. Continents were used to classify the data comparison, which has the advantages of being intuitive, allowing for more obvious comparisons between and within groups. B[a]p, PAH4, and PAH_total_ are counted as the three main reference indicators. ILCR is also considered as the representative of health outcomes. The difference between Nigeria (Africa) and other regions is obvious. Just for the daily intake of fish and meat products, the average value of B[a]p in Nigeria came to 81.2 μg/day, which was approximately 325 times that of the UK. Pakistan came in second with a gap of 3.004 μg/da but still around 12 times that of the UK. In fishery products, South Koreas’ B[a]p was surprising low at only 7.77 × 10^−5^ μg/day; PAH4 was only 4.14 × 10^−4^ μg/day, which was only 1/10,000 of other data. In another study in Beijing in 2015 [33], the average daily intake of B[a]p by local residents was 0.179 μg/day, but PAH_16_ was detected as high as 18.270 μg/day. This might be due to the smog hitting Beijing at the time, which reduced the quality of food. Similarly, a study in Catalonia, Spain 2008 showed the same difference. Regardless of the distractions, the difference is still very significant in this case.

It has been pointed out above that smoking is the traditional food processing craft in both Catalonia and Croatia, which can explain why the PAH_total_ intake of Catalonia was higher than that of other European countries in 2003 and 2008. For the period between 2003 to 2008, the daily B[a]p intake showed a downward trend, but the PAH_total_ intake was gradually increasing. In another case, Croatia could not be compared over time due to limited data. However, in the range of meat and shellfish products, the concentration and B[a]p are dramatically lower than other regions in Table 2. For Croats, their diet may be influenced by religion factors: traditional dishes such as fritule are produced by deep frying, which will be served on special days, such as Christmas Eve [34]. The contrast may be due to the standardization of the production of pickled and smoked foods and the establishment of stricter food inspection standards, thereby limiting the intake of B[a]p. The seasonality of traditional diets may also be a factor in reducing the dietary PAH intake of residents. For Latvia and France, the latter’s daily intake of B[a]p and PAH4 showed higher levels. There is less information on the traditional diet in Latvia; considering the dietary habits in France, foods for infants have extremely strict management measures for PAHs, adults prefer to choose organic plant-based foods, and the elderly have high adherence to a Mediterranean diet [35,36]. This result confirmed the relationship between dietary pattern and daily intake of PAHs to a certain extent. From another perspective, Sweden, as one of the countries that follows the Nordic diet, maintains an acknowledged healthy diet (high protein, low fat, low carbohydrates, water). However, due to the pollution from natural or anthropogenic sources in the arctic environment, which affects the water quality, PAHs are passed to marine organisms [37]. Some of the main food sources of Swedish residents are fish and shellfish in the ocean, so this may be the reason for the relatively high levels of B[a]p and PAH4 in Sweden.

Italy is leading Europe with a daily B[a]p intake of 0.562 ng/d [38]. B[a]p is a marker of PAHs; this phenomenon indicates that the PAH_total_ in this region remains at a high level. It has been shown that the chrysene (Chr) level in raw milk products is high in the Augusta–Melilli–Priolo region [39], which has an important impact on the daily intake of PAHs in the local population. The Artisan diary has played a substantial role in bolstering the economic development and dietary customs of the local region, becoming a hallmark of the area. Aside from direct consumption, PAHs created during production and processing will also provide additional intake for practitioners. According to the research of Calogero and her colleagues, the concentration of Chr in cow milk is 12.56 ± 19.17, and that of goat milk is 9.25 ± 28.49 (ng/g). The average value of cow milk is higher than that of sheep and goat milk, but the standard deviation (SD) of sheep and goat milk is significantly higher, indicating that it may be affected by more factors [39]. Raw-milk cheeses in Italy, such as Bocconcini and Parmigiano Reggiano, not only attract tourists but also find their way into the diets of locals. According to research, the estimated daily intake (EDI) of raw milk products of local people is 1.75–22.96 ng/day [40]. Combined with the data of Chr, it is clear that the high PAH concentration in raw milk is closely associated with the residents’ daily PAHs intake.

In this case, ILCR is more likely to be used by researchers targeting Asian regions. After excluding limited data such as in Shandong, China (2018) and Korea (2022), it is prominent that the data from Taiyuan, China, 2014 in both cooked food and raw food are significantly greater than other cities by over 10 times, approximately. Compared with data of previous years (2010) in Taiyuan, although PAHs had an impact on ILCR figure, more undetected factors also played an essential role in the superhigh ILCR in this area.

**Table 2 healthcare-11-01958-t002:** PAHs intake (μg/day) from diet by continent and country.

	Location	Year	Daily Intake			Comments	ILCR	Reference
			benzo[a]pyrene	PAH4 ^d^	PAHs			
Europe	France	2012	0.021 ^b^	0.104				[41]
	France	1993		0.066		Only female		[42]
	Catalonia, Spain	2008	0.089 ^b^		10.140			[43]
	Catalonia, Spain	2003	0.110 ^b^		8.400 *			[44]
	Athens, Greece	2022	0.044 ^c^		6.671 *			[45]
	UK	1983	0.250		3.700			[46]
	Croatia	2019	0.014	0.060		Total meat and shellfish products		[47]
	Latvia	2015	0.023 ^b^	0.203 ^b^		Only in smoked meat products		[48]
	Sweden	2014	0.049	0.270				[9]
	Italy	2017	0.562 ^b^					[38]
Asia	Shandong, China	2018			5.236	Only in fried meat	3.75 × 10^−6^	[49]
	Beijing, China	2015	0.179 ^b^		18.270 *		3.37 × 10^−5 b^	[33]
	Taiyuan, China	2014	0.002	0.007	0.061		906.95 × 10^−6^ (cooked food) 111.51 × 10^−6^ (Raw food)	[21]
	Taiyuan, China	2010	0.572 ^e^		5.841 ^b^		4.04 × 10^− 5^ (Male) 3.87 × 10^− 5^ (Female)	[50]
	Korea	2022	7.77 × 10^−5 f^	4.14 × 10^−4 f^	1.70 × 10^−5 f^	Only in fishery products	2.60 × 10^−6^	[51]
	Pakistani	2021	3.004				2.78 × 10^−5^	[15]
Africa	Southern Nigeria	2017			0.0523	Only by fish		[52]
	Nigeria	2021	81.2 ^b^	343	1330 *	Only by meat and fish		[20]

^a^ Data in ng/kg wb/day were converted to ng/day by assuming an average body weight for adults of 70 kg. ^b^ Mean value. ^c^ UB. ^d^ PAH4. The sum of benzo(a)pyrene—BaP, benzo(a)anthracene—BaA, benzo(b)fluoranthene—BbF, and chrysene—CHR. ^e^ Median value. ^f^ μg-TEQBaP/kg/day. * Naphthalene, acenaphthylene, acenaphthene, fluorene, phenanthrene, anthracene, fluoranthene, pyrene, benz[a]anthracene, chrysene, benzo[b]fluoranthene, benzo[k]fluoranthene, benzo[a]pyrene, and ibenz[a,h]anthracene, benzo[g,h,i]perylene, and indeno[1,2,3-c,d]pyrene.

## 4. Health Risk from Dietary Exposure to PAHs

Relevant research conducted by R. Standing and his associates have shown that several metabolites produced from PAHs may trigger gene alterations and increase the incidence of lung cancer [53]. Studies have identified an array of PAHs as relative to human gastrointestinal, breast, and liver cancer [53,54,55]. Zhang and her colleagues found that large amounts of human exposure to exogenous heat-source PAHs significantly increased the incidence of esophageal cancer in the Handan-Xingtai district, China [56]. Similarly, workers with extended exposure to mixtures containing PAHs also have a heightened risk of skin, lung, bladder, and gastrointestinal cancers [57]. Meanwhile, processing and cooking techniques are also justified to be beneficial to the production of PAHs [58]. Schwingshackl and his coworkers discovered that red and processed meat was positively associated with colorectal cancer (CRC) risk [59]. Another study in 2016 mentioned that, for men, the incidence of lung cancer is significantly higher in Asia than in other regions, and the incidence of prostate cancer is higher in Europe, the Americas, and Africa. These findings can justify the impact of PHAs on the increasing human cancer rate.

### 4.1. Hot Processed Food

The extent of processing and the method significantly affect the accumulation of PAHs in hot-processed food. Surprisingly, coffee intake also plays a crucial role in the accumulation of PAHs. Roasting is a high temperature treatment that allows the flavor of coffee beans to come out better. According to Tfouni’s findings, the average PAHs level in boiling light roasts, medium roasts, and dark roasts increased steadily. However, if you use a filter to process multiple roast levels of coffee, the PAH_total_ of medium roast will be on the top. B[b]F concentrations accounted for 48.3% of the PAH_total_ identified in extensively roasted coffee and were detectable in 94% of the brewed coffee samples [60]. These findings suggest that B[b]F is a major PAH formed during coffee roasting, and it is one of the PAHs with the highest Incremental ILCR value [61].

### 4.2. Oil and Fat

Research by Yousefi and his associates pointed out that, on the ILCR index, frying oil is higher than blended oil and corn oil in Iran. Universally speaking, fluorene accounted for the largest proportion of PAH_16_ in 85% of detected frying oil brands. The average of PAH_16_ approached approximately 27.8 ng/g, four of which were above 30 ug/g. Regarding sunflower oil, corn oil, and canola oil, one corn oil brands whose concentration was over 140 ng/g was excluded. On average, sunflower oil is around 20 ng/g and corn oil is around 30 ng/g [62]. The primary factor that resulted in this outcome is likely attributed to the frying process, which introduced additional risk factors that increased the production of PAHs. It is noteworthy that lung cancer is recognized by the World Health Organization as the foremost cause of cancer-related deaths. Further, it has been established that fluorene serves as a valuable biomarker for a heightened risk of lung cancer, as evidenced by observations of DNA hypomethylation [63]. This symptom can also be caused by another epigenetic marker 5hmC, which is a critical gene for embryonic development with susceptibility to PAHs exposure. According to research by Huang and his associates, malformation fetal neural tube defects (NTDs) can also be the result of PAH overexposure. In 62 samples of NTD patients, the concentration of high-molecular-weight PAHs (H_PAHs) is inversely proportional to 5hmC%. A decrease in 5hmC% is one of the risk factors for NTD; therefore, increased PAH exposure can be directly related to the uprising risk of NTD [64].

The breast is particularly susceptible to aromatic carcinogenesis since it is mainly composed of adipocytes. However, the occurrence of breast cancer (BC) may be related to several factors [65]. For women, estrogen disturbances can stimulate the abnormal development of breast cells, the, endometrium, and ovaries. The study by Alexandra and her colleagues highlights the effect of PAHs on the increasing incidence of BC. Grilled and smoked meat exposures accounted for a significant portion of total exposure in women with BC. In Long Island, New York more than 70% of the affected women took more than one serving of grilled and smoked meat per week, and 69.9% of the population consumed more than 55 servings of smoked barbecue meat per year. This group of people is usually classified as nonlow PAH exposure. In the survey sample, the 95% credible intervals and odds ratios (OR) of diet reached 1.13, second only to the leading exposure source of vehicular traffic [66].

## 5. Conclusions

This article reviews the content of PAHs in food, the daily intake of PAHs in humans, their dietary characteristics, their relationship with the human daily intake of PAHs, and their impact on cancer risk. The ingestion of food-source PAHs occurs through the gastrointestinal tract and is stored in adipose tissue. Due to its fat solubility and lipophilicity, rendering their elimination is an arduous task. Therefore, the prevention of PAHs is critical. Sources of PAHs present in food include soil, air, marinades, cooking techniques, external factors during smoking, and food packaging [67]. To reduce the indirect impact on food products, it is imperative to impose more rigorous regulations pertaining to food safety in regions where elevated levels of PAH contamination have been identified, and to eradicate environmental PAHs. For regions that prefer heavy oil and salt, the most efficient way to prevent PAHs may be enhancing education on healthy diet concepts; using more steaming and stewing methods instead of smoking, frying, and grilling; reducing the frequency and quantity of fried and fat-rich foods to avoid excess amount consumption.

As for further reducing the content of PAHs in food items, more attention should be paid to processing methods since they are a critical factor in the formation of these compounds. Grilling and smoking are cited as two prominent methods that increase PAH intake. For regions which have significant coastal areas and seafood production, methods that involve less smoking and frying can be selected when processing and handling seafood to reduce excessive PAHs produced during pickling or preservation. For countries where foods are frequently preserved by smoking and grilling, the optimization of traditional food processing technology is also crucial. Implementing proper food processing techniques by replacing charcoal, optimizing the method and time of connecting between smoke and food items, etc., can avoid health hazards related to PAHs exposure [67,68]. Overall, when residents in different regions choose food, PAH content resulting from food processing methods should be a consideration when selecting foods. Choosing organic vegetables and livestock from less polluted regions and ensuring thorough washing can also contribute to minimizing dietary PAH intake. While approaching the target of reducing PAH intake, reductions in the incidence of cardiovascular diseases and cancer are also expected. In order to improve residents’ quality of life, health education is a crucial tool for raising inhabitants’ consciousness and attention.

## Figures and Tables

**Figure 1 healthcare-11-01958-f001:**
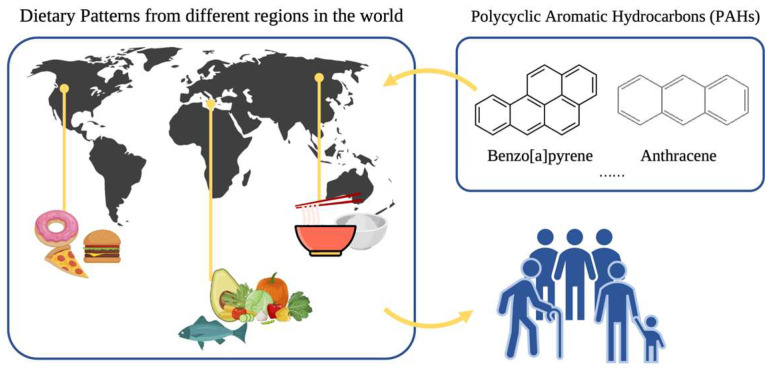
PAHs exposure from different dietary patterns.

**Table 1 healthcare-11-01958-t001:** PAHs concentrations in the daily food from different continents, countries, and regions.

Year	Area	Food Item	PAH_16_ (ng g^−1^)	Reference
2019	China (southwest)	TDS ^1^	12.04 ± 8.52	[13]
2021	Pakistan (northwestern)	Vegetables	103.6	[15]
		Fruits	132.8	
		Fish	242.8	
		Chicken	171.5	
		Milk	158.2	
		Egg	121.4	
		Wheat	168.64	
		Maize	158.88	
		Rice	52.60	
2019	Sub-Saharan Africa	Smoked fish	310.1–310.2	[16]
		Peanut oil	14.4–14.8	
		Other vegetable oil	25.0–25.3	
		Palm oil	22.6–22.7	
		Other nuts/seeds	20.8–21.0	
		Other fat/oil	2.3–2.6	
		Palm nut	5.7–5.8	
		Chili/pepper	4.6–4.7	
		Cassava dry	0.8–1.0	
		Broth/bouillon cube	0.5–1	
		Peanuts	0.4–0.8	
		Sugar	0–0.6	
		Yam (dry)	0.3–0.4	
		Concentrated/dehydrated milk	0.1–0.6	
		Tam (fresh)	0–0.2	
		Potato (fresh)	0–0.2	
2018	Shanghai, China	Shanghai green cabbage	238.9	[17]
		Chinese cabbage	260.6	
		Romaine	320.6	
		Broad bean	126.9	
		Lettuce	204.6	
		Daikon	78.9	
2013	Shenzhen, China	Vegetable	12.8	[18]
		Pork	66.5	
		Rice	47.1	
2008	Spain (12 cities)	Meat and meat products	38.99	[19]
		Fish and shellfish	2.87	
		Vegetables	1.22	
		Tubers	0.73	
		Fruits	0.81	
		Eggs	3.62	
		Milk and dairy products	8.04	
		Cereals	1.27	
		Oil and fat	18.75	
		Industrial bakery	1.43	
2021	Khana	Beef	16.7	[20]
		Mutton	8.06	
		Fish	6.42	
		Chicken	25.4	
	Choba	Beef	8.06	
		Mutton	5.78	
		Fish	6.56	
		Chicken	7.42	
2014	Taiyuan, China	Raw food	93.11	[21]

^1^ TDS: total diet study [22].

## Data Availability

No new data were created.

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
