# Peer review of "Recent Evidence on Polycyclic Aromatic Hydrocarbon Exposure"

_healthcare, 2023, doi:10.3390/healthcare11131958_

Round 1

Reviewer 1 Report

The authors are to be congratulated for their high-quality narrative review that provides a comprehensive conclusion of the relationship between the intake of 11 various polycyclic aromatic hydrocarbons (PAHs) and different dietary patterns. The manuscript is well written and easy to follow. Nonetheless, I would like to see the following minor issues to be addressed:

1. Please include the methodology using the guidelines in Systematic Reviews in Health Care: Meta-Analysis in Context, 2nd Edition by Egges et al. which can be accessed using the following link https://www.wiley.com/en-us/Systematic+Reviews+in+Health+Care%3A+Meta+Analysis+in+Context%2C+2nd+Edition-p-9780727914880

1. Please rephrase the following statement to emphasize the concentrations of PAHs “Generally speaking, the average level of vegetables in Shanghai (205.1 ng/g) showed a significantly higher than any other information below. In the perspective of process food, smoked fish in Sub-Saharan Africa was far ahead by approximately 310.1 ng/g.”

2.  In table 1, please change “Pakistani” to “Pakistan” and specify the number of cites as the case for Spain. Also please replace “cow meat” with “beef”, “goat meat” with “mutton” and “chicken meat” with “chicken”.

3. Please define the following acronyms: CAC and RCP on first mention.

4. Introducing a heat map with the highest level of PAHs by country and products would be very interesting. Please try to construct it.

Careful re-reading by the authors is recommended to take care of some minor editorial blemishes including grammar, punctuation, spelling, space, misplaced words and improvement of overall readability.

Reviewer 2 Report

The article provides evidence of PAH exposure due to different region and culturally specific dietary patterns.  The authors highlight the importance of acknowledging stark differences in dietary patterns from within the same country, and these differences could result in vastly different exposures.  However, the authors' plea for more propaganda to prevent excessive intake of PAHs might be strengthened if a direct correlation could be shown with respect to the dietary patterns and relevant health outcomes.  Only at lines 303-307 did the authors allude to a specific correlation.  To address this, the authors might consider revising Table 2 to include prevalence of relevant health outcomes for each of the regions indicated. Also, since dietary patterns are the focus of this study, the authors might also consider if there are any dietary patterns that show protective effects despite exposure to the PAH. 

Other minor comments:

line 36-38, please provide a citation following the sentence beginning, "Relevant research conducted by R Standing..."

B[a]p is not 'defined' until line 198. It should be defined at 1st mention of benzo[a]pyrene.

lines 68-69, 79-80, 83-84 seem overstated given that direct correlations to adverse health outcomes are provided here.

minor word choice or sentence structures can be addressed at:

line 12: "with pointed'

line 48: "on behalf of people began"

line 59: "To food"

lines 66-68 can be revised to read, "Based on the average of the two studies conducted between the years 1999-2010, the B[a]p content of ...."

line 90: "To divide"

Reviewer 3 Report

In this review, authors have described a more recent understanding of PAH exposure from diet and regional differences in PAH exposure based on their dietary patterns. Overall, the authors have provided the content to highlighted the problem of PAH exposure but the content needs to be better organized and the text should be reviewed by for professional English language editors to make it more comprehensible to readers.

I have attached the PDF file with my comments.

Please see above.

Round 2

Reviewer 2 Report

Authors have addressed reviewer comments.

Authors have addressed reviewer comments.

Author Response

Thanks for your affirmation. We would like to express our gratitude again for your help to improve our work.

Reviewer 3 Report

Dear authors.

Thank you for making the corrections. I had a few more minor corrections and suggestion in the attached PDF. Proofreading for English language would improve the readability.

Thank you.

 Proofreading for English language would improve the readability.
